# Neurotransmitters in Type 2 Diabetes and the Control of Systemic and Central Energy Balance

**DOI:** 10.3390/metabo13030384

**Published:** 2023-03-04

**Authors:** Amnah Al-Sayyar, Maha M. Hammad, Michayla R. Williams, Mohammed Al-Onaizi, Jehad Abubaker, Fawaz Alzaid

**Affiliations:** 1Dasman Diabetes Institute, Kuwait City 15462, Kuwait; 2Department of Anatomy, Faculty of Medicine, Kuwait University, Kuwait City 13110, Kuwait; 3Institut Necker Enfants Malades-INEM, Université Paris Cité, CNRS, INSERM, F-75015 Paris, France

**Keywords:** neurotransmitters, energy balance, diabetes, metabolism, obesity

## Abstract

Efficient signal transduction is important in maintaining the function of the nervous system across tissues. An intact neurotransmission process can regulate energy balance through proper communication between neurons and peripheral organs. This ensures that the right neural circuits are activated in the brain to modulate cellular energy homeostasis and systemic metabolic function. Alterations in neurotransmitters secretion can lead to imbalances in appetite, glucose metabolism, sleep, and thermogenesis. Dysregulation in dietary intake is also associated with disruption in neurotransmission and can trigger the onset of type 2 diabetes (T2D) and obesity. In this review, we highlight the various roles of neurotransmitters in regulating energy balance at the systemic level and in the central nervous system. We also address the link between neurotransmission imbalance and the development of T2D as well as perspectives across the fields of neuroscience and metabolism research.

## 1. Introduction

Diabetes mellitus is a chronic metabolic disease characterized by hyperglycemia that arises due to insufficient insulin production from the pancreas or from inadequate insulin utilization by the body, or a combination of both [1]. Approximately 5% of patients with diabetes suffer from Type 1 Diabetes (T1D), an autoimmune condition in which patients rely on exogenous insulin. The remaining majority have Type 2 Diabetes Mellitus (T2D), characterized by insulin resistance in its early stages. The mechanisms of disease progression in T2D, and other minor frequency forms of diabetes are under active investigation [2]. The International Diabetes Federation (IDF) indicated that 537 million adults globally were living with diabetes in 2021, with an expected 45.8% increase in the number of cases by 2045 [2].

T2D often goes undiagnosed until complications arise, and screening for the disease typically occurs only in obese patients [3]. The association between obesity and T2D is strong, yet some obese individuals maintain a healthy metabolism [3,4]. Therefore, only some mechanisms of insulin resistance are dependent on obesity. A chronic positive energy balance leads to obesity, which increases the risk of developing insulin resistance and T2D [5]. Positive energy balance results from either increased energy intake due to high calorie consumption or decreased energy expenditure due to a lack of physical activity, or a combination of the two. Readjusting energy balance to reduce body weight can improve diabetes. In fact, a 10% reduction in weight has been reported to reverse clinical diagnosis of T2D in some patients with obesity [6].

The basic mechanism for obesity is excess glucose and insulin; glucose stimulates insulin release allowing its utilization by the muscles and storage of the excess as fat. Multiple in vitro and in vivo studies have demonstrated that insulin resistance is part of the pathogenesis of T2D as it compromises efficient use of glucose [7,8,9,10]. This also leads to elevated glucagon levels and glucose production by the liver [11,12]. Persistent high glucose levels are concomitant with the stimulation of fatty acid (FA) release from peripheral storage tissues (i.e., adipose tissue), attempting to provide an alternative energy substrate. However, as the disease progress FAs fail to be utilized efficiently as an energy substrate, they are stored ectopically, and this prevents the compensatory mechanism of increased insulin production [13]. Moreover, associated with T2D are increased markers of chronic low-grade systemic inflammation as well as an altered gut microbiome and multi-organ pathologies [14,15].

Pioneering work by Claude Bernard in the 19th Century linked the brain to systemic glucose homeostasis in which a needle was used to stimulate a region of the brain in dogs, inducing a temporary diabetic state in the animal [16]. In the 1960s, insulin was found to be present in the Central Nervous System (CNS), and rather than being only of pancreatic origin, it was hypothesized and confirmed that the brain can synthesize its own insulin [17,18]. Insulin receptors were also found on neurons in many brain regions, with the highest density in the olfactory bulb and hypothalamus [19,20]. Functional insulin signaling in the brain via the PI3K/Akt and Ras/Raf/MAP pathways is metabolically protective, neuroprotective, and has a positive effect on neuroplasticity [21]. Disruption of insulin action in the brain alters neural and glial cell function at the synaptic level [22,23] and is associated with neurodegenerative and cognitive disorders as well as psychiatric diseases [24,25,26]. Experimentally, the selective disruption of insulin receptors in the brain also leads to reversible diet-induced obesity and peripheral insulin resistance [27,28,29,30]. Multiple studies have also reported an evident link between insulin-resistant peripheral tissues and the CNS (e.g., gut–brain axis, liver–brain axis, and central leptin resistance) [31,32]. Such phenomena are gaining attention for potential roles in systemic insulin resistance that is developed in obesity and T2D [32,33].

In this review we discuss how metabolic disturbance, in the form of diabetes, affects neurological functions with a focus on energy balance and neurotransmission. Neurotransmitters are a class of communication molecules that ensure normal nervous system function, interacting systemically and with various tissue microenvironments. Reviewed below are the ways in which neurotransmitters and neurological functions are affected in metabolic disease. To our knowledge, no recent in-depth reviews have addressed how systemic dysmetabolism mechanistically alters neurotransmission and the ways this contributes to increased risk of neurodegenerative diseases in patients with diabetes.

## 2. Type 2 Diabetes and Neurological Complications

Several longitudinal studies show that patients with diabetes are more susceptible to cognitive impairment [34,35,36]. Altered brain function and metabolism is associated with insulin resistance and with vascular complications, dyslipidemia, and hypertension, which are all common in T2D [37,38]. A population-based study also showed that older patients with diabetes had higher risk of developing Alzheimer’s disease (AD), linking diabetes to age-related neurodegeneration [39]. From the above studies and other mechanistic investigations, it is clear that impairment of glucose metabolism in the context of diabetes is a key element in the onset and progression of AD [40,41]. In cases of hyperglycemia, reactive oxygen species (ROS) production is elevated through stimulation of the polyol pathway. This leads to formation of advanced glycation end products (AGE) and causes considerable oxidative stress [42]. Accumulation of AGEs contributes to the pathological aspects of neurodegenerative diseases including AD, Parkinson’s, and Huntington’s diseases [43]. Kong et al., investigated the role of AGEs in AD and diabetes in vivo and found that mice injected with AGEs exhibited symptoms of AD with impaired memory and increased levels of amyloid precursor proteins (APPs) and tau [44]. Another study also highlighted that AGE receptor (RAGE) contributes to the decrease in locomotor activity and spatial memory in streptozotocin (STZ)-induced hyperglycemia in mice [45]. Another mechanism that links diabetes to cognitive decline is inflammation. It is well-established that diabetes and its complications and comorbidities occur on a background of low-grade chronic inflammation, which participate in neurodegenerative processes that impair cognition, synaptic plasticity, and neurotransmission [38,46]. In particular, decline in executive functions and psychomotor speed in patients with T2D are associated with alterations in neurotransmitter release, neuronal dysfunction, and neurodegenerative damage [47].

## 3. Neurotransmitters

Neurotransmitters are a massive family of chemical messengers (>100 known neurotransmitters) involved in synaptic transmission; they regulate physiological functions in the central and peripheral nervous system [48]. They can be classified based on their physiological functions (excitatory or inhibitory), mode of action, molecular structure, or chemical group (Table 1) [49]. Neurotransmitters are generally stored in vesicles at the axon terminal of presynaptic neurons and can be released in response to action potentials, a rapid rise and fall in voltage, or membrane potential across the cellular membrane. Once released, they diffuse across the synaptic cleft to act on receptors on postsynaptic neurons to exert their effects (Figure 1A) [50]. This process creates neuronal circuits that facilitate communication between neurons through chemical synapses. Abnormal neurotransmitter levels reflect dysregulation of brain functions that manifest as physical, psychologic, and neurodegenerative diseases [51,52,53,54,55].

Glutamate, Gamma-aminobutyric acid (GABA), dopamine, serotonin, norepinephrine, and acetylcholine are of clinical relevance and are the most studied neurotransmitters. Most of the synaptic activity and intracellular signaling in the brain is accounted for by glutamate (excitatory) and GABA (inhibitory) [56]. Healthy function of neurotransmission requires a balance between excitatory and inhibitory (E/I) signals that is crucial for proper neuronal firing and synaptic transmission. This is mediated by glutamate through the activation of the ionotropic receptors N-methyl-D-aspartate (NMDA), α-amino-3-hydroxy-5-methyl-4-isoxazole propionic acid (AMPA), and the metabotropic glutamate receptors (mGluRs); and by GABA through GABA_A_ and GABA_B_ receptors (Figure 1B). Appropriate activation of these receptors ensures that the right neuronal excitability level is translated to different brain regions [57]. Thus, a balanced interaction between glutamate and GABA ensures physiological homeostasis, and alterations in E/I balance leads to disorders such as epilepsy and AD [58,59,60]. A similar concept also applies to the development of other chronic conditions such as Parkinson’s (acetylcholine/dopamine) and Huntington’s (glutamate/dopamine) diseases [61,62].

**Table 1 metabolites-13-00384-t001:** Neurotransmitter classification by chemical group.

Category	Neurotransmitter	Function
Amino Acids	Gamma-aminobutyric acid (GABA)	Learning, memory, locomotion, metabolism mediators, appetite regulation [63,64]
	Glutamate (Glu)	Memory, learning, cognition, appetite regulation [48,64]
	Glycine (Gly)	Motor control, sensory and auditory processing, cardiovascular, and respiratory functions [65]
Amines	Dopamine (DA)	Motivation, memory, attention, locomotion control [66]
	Norepinephrine (NE)	Emotional arousal, regulating blood pressure, mood, appetite [49]
	Epinephrine	Boosts oxygen and glucose supply to brain and muscles, increases awareness [48]
	Serotonin (5-HT)	Regulate sleep–wake cycle, mood, appetite and digestion [65]
	Histamine	Regulate sleep–wake cycle, stress response, appetite and memory [67]
Acetyl Choline	Acetylcholine (Ach)	Cognition, learning, memory, modulation of electrical, and mechanical functions of the heart [51,68]
Other	Nitric Oxide (NO)	Learning, memory, homeostatic functions [69]
	Hydrogen Sulfide (HS)	Neuromodulator, smooth muscle relaxation [48]
	Purines (ATP)	Controls intracellular energy homoeostasis, autonomic control, sensory transduction [70,71]

Neurotransmission also occurs through electrical synapses that contain intercellular aggregate channels known as gap junctions (GJs) allowing electrical and metabolic communication between adjacent neurons. GJs are formed by hemichannels from each side of the synapse which are composed of transmembrane proteins called connexins that allow the transfer of ions and small molecules between neurons [72,73]. Cx36 is the most abundant connexin type in neurons and it forms most of the electrical synapses in the CNS [74]. Connexin GJs play a homeostatic role in CNS physiology. This includes synaptogenesis, neuronal differentiation and circuit formation and maturation [73]. GJs also regulate neural activity oscillations (i.e., maintaining a synchronized excitatory and inhibitory electrical activity) that allow robust communication between neuronal assemblies. Alterations in connexin GJ activities can impact their expression and function leading to the progression of neurodegenerative diseases, including AD and Parkinson’s disease and epilepsy [75,76,77,78].

## 4. Effect of Metabolic Dysregulation on Neurotransmitters Functions

T2D often exists with other factors that disrupt homeostasis and worsen metabolic dysregulation. For example, obesity is the result of chronic energy imbalance and is associated with low-grade chronic inflammation that strongly influences the diabetic state and susceptibility to complications and comorbidities, such as cardiovascular or liver disease [37,79]. Recent evidence suggests that cognitive impairment and dementia are emerging complications of T2D and obesity, and that this outcome is associated with alterations in neurotransmitter homeostasis and synaptic activity [80,81,82]. Peripherally, glucotoxicity has been reported to increase glutamate levels, inducing β-cell dysfunction and neuronal injury through the activation of pancreatic NMDA receptors; and blocking these receptors improved β-cell function in vitro and in vivo [83]. D’Almeida and colleagues reported that metabolic alterations in T2D dysregulate the E/I balance between glutamate and GABA causing cognitive impairment [84]. Changes in amino acid neurotransmitter homeostasis was also observed in obese and diabetic rats as a result of impaired brain glucose metabolism [81]. Furthermore, hyperglycemic conditions lead to GJ impairment that disrupts astrocyte–neuron communication leading to changes in brain function [85]. Head et al. reported that loss of Cx36 GJs disrupted glucose homeostasis through the alteration of oscillating insulin levels in mice [86]. This was further confirmed in patients with diabetes, as they exhibited disruption in Cx36 GJ permeability and Ca^2+^ electrical activity. Treatment with Modafinil restored Cx36 GJ function, maintained cell viability, and protected against β-cell dysfunction [87].

In obesity, excessive intake of macronutrients such as carbohydrates and fats lead to failure in homeostatic mechanisms that are involved in regulating energy balance, especially in the hypothalamus [88]. Exposure to obesogenic diets leads to early hypothalamic alterations indicating a causal role of central dysregulation in the onset of obesity [37]. Functional dysregulation of neurotransmitters such as dopamine disrupts the hypothalamic circuitry controlling satiety, leading to uncontrolled weight gain [89,90]. These studies highlighted that hyperglycemia and/or dyslipidemia significantly impact hypothalamic neurotransmitter function and the neurotransmission process, affecting systemic energy balance and metabolism.

## 5. Role of The Hypothalamus in Controlling Energy Balance

Located in the base of the brain, the hypothalamus plays an essential role in regulating energy balance through the integration of neural circuits and humoral factors (e.g., hormones) in response to food intake and energy expenditure [91]. It consists of several nuclei with different functions that communicate systemically with one another through neurotransmitters and neuropeptides (Table 2, Figure 2) [92]. In particular, the Arcuate Nucleus (ARN) is a major contributor in regulating food intake and energy expenditure via its two key neuronal subpopulations: orexigenic neuropeptide Y (NPY)/agouti-related protein (AgRP)-expressing neurons and the anorexigenic proopiomelanocortin (POMC)/cocaine and amphetamine-regulated transcript (CART)-expressing neurons. The NPY/AgRP and POMC/CART-expressing neurons control energy balance by promoting and suppressing appetite, respectively [92]. Successful function of the hypothalamus is dependent on two major factors, locally efficient neurotransmission, and the integration of systemic signals.

### 5.1. Hypothalamic Neurotransmitters

As the hypothalamus undergoes rapid changes in circuit connectivity and function, it is important to sustain a balanced neurotransmission process between the hypothalamic neurons to maintain appropriate behavior and endocrine functions [94]. The hypothalamus is considered a primary site for appetite regulation in which orexigenic neurons (NPY/AgRP) and anorexigenic neurons (POMC/CART) regulate appetite. Amino acid neurotransmitters glutamate and GABA account for most of the synaptic activity in the hypothalamus [64]. As such, glutamatergic and GABAergic phenotypic markers, for example VGLUT2, have been reported in POMC+ neurons; glutamatergic innervation has also been reported in NYP+ and POMC+ neurons in the ARN [95,96]. These studies indicate the presence of a dense plexus of glutamatergic fibers in the hypothalamus. Interestingly, studies have also demonstrated that POMC+ neurons exhibit a level of plasticity in their expression of glutamatergic and GABAergic markers. Such plasticity promotes restoration of energy homeostasis in models of genetic interference [97]. Moreover, Trotta et al. reported that GABAergic-POMC neurons regulate food intake and energy balance through the DMN-NPY pathway in vivo [98], and several other studies have shown that GABA signaling is required in AgRP+ neurons to stimulate and maintain feeding behavior. However, the role of GABA release from POMC neurons remains unclear [99,100,101]. These studies indicate that the synaptic release of amino acid neurotransmitters by the hypothalamic ARN neurons has a significant role in modulating energy balance.

Monoamines, such as dopamine and serotonin, also play a role in appetite regulation. A study by Zhang et al. described the role of novel tyrosine hydroxylase (TH)-expressing neurons in the ARN and found that light-mediated optogenetic stimulation resulted in dopamine release [102]. This increases food intake by inhibiting POMC+ neurons and stimulating NPY/AgRP+ neurons [102]. Dopamine secretion initiates feeding patterns (i.e., meal size, frequency, and duration) and regulates food intake in the VMN and LHA [103]. To balance this, serotonin is released in the LHA during enhanced feeding, promoting satiety [103]. Few studies have investigated the role of acetylcholine in the hypothalamus. Jeong et al. found that increased DMN cholinergic neuronal activity regulated food intake through ARC POMC neurons in vivo [104]. Others suggest acetylcholine modulates appetite suppression, and under fasting conditions, elevated cholinergic activity is observed in hypothalamic regions due to the anticipation of food intake [105,106].

### 5.2. Central Melanocortin System

The central melanocortin system is one of the most important systems in the regulation of appetite and energy homeostasis. It consists of neurons that produce endogenous melanocortins (melanocortin neurons) and the downstream neurons that express melanocortin receptors. Melanocortin neurons are POMC and AgRP neurons [107]. POMC is a preprohormone that undergoes post-translational processing (proteolytic cleavage) to produce the melanocyte stimulating hormone (MSH—α, β and γ). These peptides can bind to and activate the melanocortin receptors (MCRs). MCRs belong to the G protein-coupled receptor (GPCRs) family and five members have been characterized to date (MC1R to MC5R). These receptors were shown to be expressed in several tissues and therefore have distinct physiological functions [108]. MC1R is mainly expressed in the skin and hair follicles and are known to regulate melanogenesis; MC2R is expressed in the adrenal cortex and is considered the classical ACTH receptor; MC3R and MC4R are neural receptors as they are present at high levels in the CNS and they play a role in mediating energy homeostasis; and MC5R has a broad tissue distribution especially in exocrine glands [109]. A unique aspect about MCRs compared with other GPCRs is the existence of the endogenous antagonist AgRP which can bind to the receptors to prevent the binding of MSH [110]. Interestingly and despite being originally classified as a competitive antagonist, AgRP was later shown to have the capacity to activate different signaling pathways downstream MCRs and act as a biased agonist [111].

Similar to other members in the GPCRs family, the central melanocortin receptors can activate several signaling pathways. These receptors can couple to the three main heterotrimeric G proteins, Gs, Gi, and Gq [112]. Coupling to the stimulatory G protein subunit (Gs) can activate Adenylyl Cyclase (AC) resulting in the release of Cyclic Adenosine Monophosphate (cAMP) and activation of Protein kinase A (PKA). Coupling to Gi on the other hand leads to inhibiting AC. Studies have also showed that these receptors can couple to Gq which results in the activation Phospholipase C (PLC) leading to the hydrolysis of phosphatidylinositol-4,5-bisphosphate (PIP2) into diacylglycerol (DAG) and inositol-1,4,5-trisphosphate (IP3). IP3 leads to the secretion of Ca^2+^ from intracellular stores while DAG activates PKC [113,114]. In addition to these G protein-dependent signaling pathways, many studies have presented evidence on the activation of a number of kinases by MC3R and MC4R including extracellular signal-regulated kinases 1/2 (ERK1/2), c-Jun N-terminal kinases (JNK), 5′AMP-activated protein kinase (AMPK), and protein kinase B (PKB or AKT) [111,114,115,116,117,118,119].

MC3R and MC4R were shown to play an important role in the regulation of glucose homeostasis and insulin sensitivity. This was first confirmed from knockout animal models that displayed elevated insulin levels and reduced insulin sensitivity in addition to the hyperphagia and obesity [120,121]. Studies also reported that treatment with the melanocortin agonists or antagonists can directly mediate insulin action [122,123]. Furthermore, patients with mutations in the MC4R gene were characterized by severe hyperinsulinemia [124]. A study also reported an increased risk of developing T2D in children and adults carrying a mutation in MC4R and this was independent of Body Mass Index (BMI) in children only [125]. However, a meta-analysis on more than 100,000 adults confirmed the significant association between MC4R rs17782313 polymorphism and increased T2D risk and this association was independent of BMI [126]. Furthermore, the central melanocortin system was shown to directly control the peripheral lipid metabolism [127]. Specifically, using an antagonist against hypothalamic melanocortin receptors resulted in an increase in energy stores by affecting several peripheral tissues. For example, in the liver, increased triglycerides and lipoprotein synthesis and secretion was observed, in addition to decreased thermogenesis and glucose uptake in the brown adipose tissue as well as in muscles. Blocking neural MCRs also increased triglycerides synthesis as well as glucose uptake and insulin sensitivity in white adipose tissue [127]. It is also worth noting that pituitary MSH plays a role in promoting glucose uptake in the muscle; however, it was found that this effect is mediated by peripheral MC5R [128].

There is a significant association between the activity of the central melanocortin system and neurodegenerative changes [129]. Studies have shown that treatment with MC4R agonists has neuroprotective effects in cerebral ischemia [130] and in the progression of AD [131,132]. MC4R activation seems to have anti-inflammatory and anti-apoptotic effects that can help reduce DNA damage, decrease neuronal loss, and reduce hippocampal injuries. It has also been suggested that melanocortins can counteract the cognitive decline in AD and other neurodegenerative disorders by reducing the level of β-amyloid peptides in the cerebral cortex and hippocampus [132]. It was also suggested that melanocortins can target multiple pathophysiological pathways up- and downstream β-amyloid and tau causing an increase in synaptic transmission and plasticity [131]. Preclinical studies in animal models generated very promising data; as treatment with MC4R agonists such as α-MSH and melanotan-II reduced food consumption and improved insulin and glucose levels [122,123,127,133]. On the contrary, when antagonists such as AgRP or SHU9119 were used, an increase in appetite and body weight was observed leading to insulin resistance [133].

Establishing a role of the melanocortin system in regulating energy homeostasis and glucose metabolism created great interest in utilizing these functions in pharmacological applications. Several ligands were developed, and some compounds reached phase I and II clinical trials. The beneficial effects varied from reduction in appetite and weight loss to improvement in glucose metabolism [134]. However, the progress was halted due to some undesirable side effects including elevated heart rate and blood pressure. Recently, a highly selective MC4R agonist called Setmelanotide was approved by the Food and Drug Administration (FDA) for the treatment of monogenic obesity [135]. Setmelanotide did not cause any cardiovascular side effects; instead it was reported that it can reduce blood pressure and improve glucose homeostasis [136,137]. It would be important to investigate the possibility of re-purposing such approved drugs or use them to design melanocortin analogs for the treatment of T2D and metabolic disorders. Combination therapy is also becoming a viable option considering the complexity of the metabolic syndrome. In fact, a study reported that co-administration of Setmelanotide with the GLP-1 receptor agonist (liraglutide) in diet-induced obese mice was more effective than monotherapy in increasing insulin sensitivity, decreasing fat mass, and improving energy expenditure [138].

## 6. Regulating CNS Energetic Demands

### 6.1. Energy Metabolism in the Brain

The ability of the human brain to carry out complex behaviors, make decisions, and process social and emotional contexts comes with a high energy demand [139]. While it covers 2% of total body weight; the brain accounts for 20% of the body’s resting metabolic rate and consumes 25% of the body’s glucose [140]. The brain’s own energy metabolism, or neuroenergetics, is maintained by the vascular supply of oxygen and glucose and adapts for localized neural activity [141,142]. It was reported that during synaptic transmission and ion influx regulation, a single resting neuron uses about 4.7 × 10^9^ Adenosine 5′-triphosphate (ATP) molecules per second, indicating the staggering amount of neuronal energy usage for these processes [143].

### 6.2. Energy Substrates: Glucose, Ketone Bodies and Lactate

In the adult brain, glucose is considered as the main energy substrate entirely oxidized to CO_2_ and H_2_O via glycolysis. The tricarboxylic acid (TCA) cycle and oxidative phosphorylation also produce ATP for energy-dependent reactions similar to other tissues [144]. Glucose transporters (GLUTs) are expressed in different parts of the CNS and allow neuronal glucose uptake from the circulation (Table 3). Glucose is imported to the cytoplasm of the cell and converted to pyruvate via glycolysis [62,65]. Under anaerobic conditions, pyruvate is converted into lactate and released to the extracellular space [145]. Lactate is considered a by-product that can dysregulate brain function if present in excessive amounts [146,147]. However, in contexts demonstrated by in vitro modelling, lactate appears to be used as an alternative substrate to sustain synaptic plasticity and neuroprotection functions during tissue development [148]. In adults, lactate has been reported to cover up to 10% of the brain’s energy requirement under steady state, leaving glucose as the main substrate [140].

Ketone bodies can also act as an energy substrate, especially in the case of diabetes or under conditions of starvation or fasting when glucose levels are low [144,149]. Physiologically, ketone bodies are generated in the liver when metabolism switches from carbohydrates to fats. Such conditions can be induced by a ketogenic diet (KD), which is a high fat and low carbohydrate diet [150]. Once ketone bodies cross the Blood–Brain Barrier (BBB), they enhance mitochondrial function, ATP generation, synaptic plasticity, and the myelination process at the early stages of brain development, making them a potential therapeutic target for neurodegenerative diseases [144,150]. Administrating KD in patients with AD showed significant improvements in cognitive and executive functions [151], Likewise, patients with Parkinson’s also had enhanced non-motor functions and cognition [152]. Furthermore, studies demonstrated that ketone bodies can reduce glutamate excitatory neurotransmission effect and the firing rates of neurons by opening potassium ATP channels and activating GABA receptors providing a therapeutic mechanism for AD and epilepsy [153,154]. Hypometabolism is often accelerated in the context of insulin resistance which destabilizes the brain network, triggering diabetes-induced dementia [150,155]. Some studies have highlight the role of KD in improving insulin sensitivity and glycemic control [150,156]. Paradoxically, patients with diabetes are more prone to diabetic ketoacidosis, which can lead to comas or death. Diabetic ketoacidosis occurs when the body does not have enough insulin to trigger the use of glucose, lipids are then used to compensate, resulting in production of ketones and their build up in blood. Despite potential benefits of ketone bodies in neurodegenerative disease, prolonged exposure to them in the hypothalamus can further dysregulate insulin secretion and energy homeostasis, leading to hyperphagia and obesity [157,158].

**Table 3 metabolites-13-00384-t003:** Glucose transporters (GLUTs) functions in the brain [159,160].

Glucose Transporter	Function
Glut1	Maintaining glucose transportation into the CNS via BBB
Glut3	Highly distributed in the brain and expressed in neurophil synapses activities
Glut4	Regulating insulin signaling in the CNS
Glut8	Supporting neuron cells glucose requirements

### 6.3. Cell-Type Specific Mechanisms

At a cellular level of the CNS, glial cells (astrocytes, oligodendrocytes, and microglia) outnumber neurons by a factor of 10. Astrocytes alone account for almost 50% of total brain volume [145]. These cells play a role in regulating neurotransmitter release, developing synaptic junctions and controlling brain energy metabolism [161]. They express neurotransmitter receptors that enable them to detect synaptic activity and provide energy substrates to active neurons. This is achieved by astrocytes surrounding brain capillaries to form a barrier for the uptake and distribution of substrates such as glucose [144]. Studies demonstrate that astrocytes exhibit high rates of glucose metabolism, as they are also the only brain cells capable of storing glycogen. A current leading hypothesis for maintaining activity in situations of low glucose and high neural firing is the formation of the astrocyte–neuron lactate shuttle (ANLS). The ANLS is formed when astrocytes sense increased neural activity from the release of glutamate at the synapse. This triggers glycolysis to produce lactate as a neuronal energy substrate (Figure 3) [147,162,163]. Furthermore, studies have reported the importance of the ANLS in long-term memory formation; and dysregulation of the lactate it supplies contributes to hypoxia-induced neurodegeneration and can also lead to stroke [164,165,166].

### 6.4. Neuronal Bioenergetics and the Regulation of Neurotransmission

Dysregulation of neuronal bioenergetics significantly impacts synaptic transmission and brain function, facilitating the progression of neurodegenerative disorders [143,167]. Because neurons are highly polarized, continuous ATP generation is required to restore action potential events that maintain synaptic transmission which lead to neurotransmitter release [168]. In addition to ATP generation, mitochondria are essential for maintaining neuronal functions and survival by regulating processes such as amino acid synthesis, Ca^2+^ homeostasis, and apoptosis [143]. Thus, perturbed mitochondrial function leads to dysregulation of neuronal bioenergetics.

The actions of the main excitatory neurotransmitter, glutamate, depend on mitochondrial oxidative phosphorylation for a maintained transport process across plasma, vesicular, and mitochondrial membranes [169]. Mitochondrial failure results in a dramatic increase in glutamate release into the synaptic cleft with a 100-fold increase in its concentration leading to excitotoxicity [170]. Furthermore, in the case of diseases such as Parkinson’s, alteration in dopaminergic neuron function has been linked to impaired mitochondrial bioenergetics since neurons increase energy demands and oxidative stress [171]. It has also been reported in AD that mitochondrial Ca^2+^ accumulation activates pyruvate dehydrogenase-(PDH) kinase and inhibits the PDH complex (PDHC). This leads to insufficient acetyl-CoA production in the cerebral cortex and hippocampus, disrupting acetylcholine synthesis [172].

Oxidative damage and mitochondrial dysfunction are key pathological processes that mediate neurodegeneration in diabetes. Abnormalities in mitochondrial function, structure, and connectivity can impact behavior and contribute to obesity [173]. Moreira et al. showed that mice under HFD exhibited reduced ATP production, altered mitochondrial dynamics and fragmentation in the hippocampus [174]. HFD also led to deficits in electron transport chain function and in oxidative phosphorylation, with reduced expression of mitochondria associated proteins such as PGC-1a and the mitochondrial transcription factor A (TFAM) in vivo [175]. Moreover, the link between insulin resistance and behavioral disorders such as anxiety and depression are associated with dysregulation of mitochondrial function and monoamine neurotransmitter homeostasis. This was demonstrated in neuron-specific insulin receptor knock-out mice that had elevated dopamine clearance and reduced mitochondrial oxidative activity [26]. Another study induced a diabetes-like phenotype in MitoPark a mouse model, which exhibits a Parkinsonian phenotype and mitochondrial dysfunction in dopamine neurons. This was observed in vivo and similar results were observed ex vivo, collectively indicating that mitochondrial dysfunction is associated with the progression of Parkinson’s disease in the context of diabetes [176].

## 7. Metabolic Factors Affecting Neurotransmission

### 7.1. Glucose Metabolism

Since glucose is the main energy substrate for the brain, disturbance in its metabolic and signaling pathways can affect its utilization and energy supply to the brain. Glucose transporters play an essential role in maintaining neuronal homeostasis; and their presence in different parts of the brain can be influenced by the organism’s diabetic status (Table 3) [159,160,177]. For example, Bussel et al. conducted an observational study and found that patients with T2D exhibited altered GABAergic neurotransmission and this was associated with lower cognitive function compared with patients without diabetes [178]. A study in a rodent model of T2D investigated the role of glycogen in supporting glutamate/GABA homeostasis in vivo; since glycogen can be used as a buffer to maintain glucose levels in the brain, use of a glycogen phosphorylase inhibitor reduced glutamate levels and increased GABA levels, disrupting neurotransmitter release [179]. Furthermore, results from another clinical study suggested that patients with diabetes had episodic memory decline which was significantly associated with abnormal amino acid neurotransmitter concentrations in the brain [180].

The dopaminergic system is also important in maintaining glucose metabolism as its activation leads to improved glucose tolerance and insulin sensitivity [181]. In the diabetic state, dysfunctional dopamine neurotransmission causes motor impairment and neurodegenerative damage [46]. Studies have also revealed that culturing dopaminergic PC12 (cells that exhibit mature features of dopaminergic neurons) under high glucose conditions lead to increased levels of reactive oxygen species (ROS) and apoptosis. This is attenuated when treating the dopaminergic neurons with the polyphenol compound resveratrol [182,183]. Diabetes also alters other monoamines in specific brain regions [184]. For example, under hyperglycemic conditions, serotonin neurotransmission is altered and decreased in the hippocampus of mice fed a high-fat diet, and this was associated with symptoms of behavioral and eating disorders [185]. Moreover, administering serotonin in vivo had a significant effect in improving glucose tolerance and utilization [186]. Another study also showed that imbalance in cholinergic neurotransmission led to cerebellar dysfunction in diabetic rats [187,188]. Further, low expression levels of nicotinic acetylcholine receptors (nAChRs) induced neuronal apoptosis in the hippocampus of diabetic mice and patients that resulted in impaired cognition [187,188].

### 7.2. Glycemic Variability

Glucose is critical for normal brain metabolism. Uncontrolled fluctuations in blood glucose levels stimulate ROS overproduction which makes patients with T2D more susceptible to microvascular complications [189]. Recent evidence suggests that fluctuation in glycemic peaks is associated with impairment in cognitive function [190,191]. A population-based study confirms this further, 5% of the study’s cohort were diagnosed with AD and this was mainly dependent on age and the duration of diabetes [192]. Quincozes-Santos et al. showed that fluctuations in glucose levels affected astrocyte function in maintaining cell proliferation and glutamatergic metabolism, inducing cytotoxicity [193]. Another study evaluated the effect of fluctuating glycemic peaks on E/I balance in T2D patients and found significant changes in glutamate–GABA neurotransmission which contributes to the pathophysiology of T2D [84].

### 7.3. Insulin Signaling

Insulin plays a crucial role in controlling blood glucose levels by facilitating cellular glucose uptake in peripheral tissues. Disturbed insulin signaling in the brain may accelerate aging, affect synaptic plasticity, and promote neurodegeneration [194]. Previous studies have reported that insulin receptors are present in different brain regions and that it is synthesized in the central nervous system with a distinct role in glucose metabolism and neuromodulation [20,195]. Decreased PI3K levels were observed in the AD brain. The PI3K-AKT pathway facilitates glucose uptake upon activation by insulin and intranasal insulin administration was found to enhance memory and cognition in vivo [196]. Reduced acetylcholine in the brain also leads to cognitive impairment in AD, and abnormalities in insulin growth factor 1 (IGF-1), gene expression results in reductions in energy metabolism and acetylcholine transferase expression which in turn increases the severity of neurodegeneration [197]. Another study also supported this concept and found that impairment of acetylcholine synthesis is due to dysregulation of insulin secretion, whereas enhanced insulin levels reverse memory loss and cognitive impairment and are associated with elevated acetylcholine concentration in the brain [198]. Moreover, Duarte et al. stated that insulin acts as a modulator of the amino acid neurotransmitters GABA and glutamate which is another reflection of its neuroprotective functions [199]. In addition, emerging evidence has been reviewed to show that the modulatory effect of insulin on GABA/glutamate is important for long-term memory consolidation, whilst deterioration of insulin signaling leads to degenerative disorders, especially in aging patients with diabetes [200].

### 7.4. Lipid Metabolism

Around 50% of the brain’s dry weight is composed of lipids making it the second organ with highest lipid content after adipose tissue under physiological conditions. Compared with adipose tissue that acts as a long-term energy store for converting FAs to triglycerides for subsequent utilization by other tissues; the brain utilizes acylated lipids primarily to generate phospholipids for cell membranes [201]. The composition of FAs varies in the brain and neural tissue and is mainly rich with polyunsaturated FAs (PUFAs) such as arachidonic acid (AA), eicosatetraenoic acid, and docosahexaenoic acid (DHA) that are critical for maintaining the nervous system’s normal function [201]. FAs can cross the BBB and be up taken by neurons and astrocytes via FA transporters such as Fatty Acid Transport Protein 1 (FATP1), Fatty Acid Transport Protein 4 (FATP4), and Cluster of Differentiation (CD)-36 [202]. Once released from the membrane, FAs are converted to a variety of bioactive mediators that enable them to participate in signal transduction [203]. Studies have showed that reduced BBB expression of Fatty Acid Binding Protein 5 (FABP5) led to lower DHA trafficking in AD, while mice lacking Fatty Acid Binding Protein 7 (FABP7) exhibited schizophrenic phenotypic traits [204,205]. This supports the fact that lipids/PUFAs play a role in regulating several brain functions, including neurotransmission, cell survival, and neuroinflammation; and dysregulation of such processes leads to various neurological disorders [203]. Literature on the effect of dietary lipids on neurotransmission processes is limited. Sandoval-Salazar et al. investigated the effect of high-fat diet on GABA levels in the frontal cortex (FC) of the rat’s hippocampus and found that high-fat diet significantly decreased GABA levels, disrupting the GABAergic inhibitory effect and role in appetite regulation [206]. In addition, Lizarbe and colleagues observed a strong switch from glutamatergic to GABAergic neurotransmission in the hypothalamus of high-fat diet-fed mice that disrupted the balance between orexigenic and anorexigenic networks, thus leading to impairment of appetite and energy homeostasis [207].

## 8. Perspectives and Future Work in the Field

The current literature was reviewed in the fields of diabetes and neuroscience. The association between the central regulation of energy balance and the development of obesity is clear: there is an increased risk of developing T2D in those patients that are obese. Loss of metabolic homeostasis in diabetes, both T2D and T1D, affects multiple organ systems, including the CNS. Indeed, patients with diabetes are at higher risk of developing CNS disorders as well as peripheral neuropathy. The development of nervous disorders is underpinned by a functional impairment. An important aspect in the function of the nervous system is appropriate neurotransmission function. All the literature cited above indicates that the neurotransmission process is affected by and can affect the susceptibility to the development of metabolic diseases. The mechanisms underlying impairments of neurotransmission and CNS function in diabetes are under active investigation, the literature has explained much of the potential mechanisms that link metabolic to neurological dysfunction. Future work in the areas described below further elucidate molecular and cellular mechanisms underlying the association between diabetes and neurological decline and importantly shed much needed on the directions of causality between metabolic and neurologic disturbance.

### 8.1. Emerging Role of Inflammation in Neurotransmitter Function

The nervous and immune systems are tightly intertwined as they modulate each other through sophisticated bidirectional crosstalk. Specifically, the brain is a highly immunologically active organ that hosts its own immune cells and allows others to circulate through its fluid filled borders in the meninges, the protective membranes that cover the CNS [208]. This is evident from multiple studies, for example knocking out immune cells from mice led to accelerated neurodegeneration that was accompanied by gliosis; while restoring these cells modulated the trophic/cytotoxic balance of glial cells and slowed disease progression [209]. Another study showed that patients with AD had a significant rise in the number of T-cells in the cerebrospinal fluid suggesting that they might have a role in AD progression [210]. On the contrary, instead of its neuroprotective role, the immune system can also cause cumulative damage to neurons due to chronic inflammatory reactions [211].

### 8.2. Neuroinflammation

Neuroinflammation refers to the accumulation of glial cells in the central nervous system in response to inflammation [211]. This occurs when astrocytes and microglia are activated immediately after injury leading to the secretion of proinflammatory cytokines (e.g., IL-1β, TNFα), cytotoxic compounds, and ROS, triggering neuronal death [211]. GABA and glutamate receptor expression is altered under neuroinflammatory conditions causing impaired spatial learning, motor function, and cognitive decline [212]. Experimentally induced inflammation through administration of poly(I:C) during gestation, dysregulated prefrontal GABAergic expression and led to the development of long-term neuropsychiatric disorders in animal models [213]. Moreover, GABA suppresses the reactive response between astrocytes and microglia to inflammatory stimuli and reduces the release of TNFα and IL-6; while an increase in TNFα levels downregulates the inhibitory synaptic strength of GABAergic neurotransmission [214,215]. Another study also looked at the effect of parthenolide, an NF-kB inhibitor, on diabetic rat models and found that the rats exhibited enhanced cognition, reduced anxiety-like behavior, and decreased TNFα and IL-6 levels in the cortex and hippocampus; which was associated with the loss of GABA and glutamate homeostasis [216].

### 8.3. Hypothalamic Inflammation

Since the hypothalamus is a major site for appetite regulation and energy balance, hypothalamic inflammation is often suggested as a cause for dysregulation of feeding behavior and body weight [217]. Obesity can, in fact, be considered as a hypothalamic inflammatory disorder resulting from the accumulation of high-fat and high glycemic index products that accumulate as ectopic fat. This eventually leads to increased BBB permeability, tanycytes damage, and activation of microglia and astrocytes pro-inflammatory pathways; which induces the secretion of cytokines such as IL-1β and TNFα alongside ROS production in the hypothalamus [218]. This is evident in patients with diabetes and obesity as a result of dysregulated energy homeostasis [218]. Moreover, obese insulin-resistant rat models of AD exhibited elevated levels of pro-inflammatory cytokines and chemokines in the hypothalamus and other brain regions suggesting a correlation between AD and hypothalamic inflammation [219].

### 8.4. Hypothalamic Regulation of Energy Homeostasis: The Roles of Sleep and Thermogenesis

Thermal and sleep regulation are also associated with the hypothalamus’ role in maintaining energy homeostasis. Previous studies indicate that AgRP and POMC stimulation promotes wakefulness and reduces sleep fragmentation, respectively [220]. The hypothalamus contains a sleep and thermal regulation center known as the ventrolateral preoptic (VLPO) area. The VLPO consists of a large population of GABAergic neurons that send signals to other brain regions to regulate sleep [220]. Disruption of GABAergic neurons in the LHA dysregulates sleep and induces vigorous eating habits. A study looked at GABAergic neurons in the DMN region of the mouse hypothalamus and found that these neurons play a role in sleep–wake changes elicited by exposure to warmth in mice, indicative of their role in maintaining sleep and energy homeostasis [221].

### 8.5. Therapeutic Approaches for Neurological Disorders

The use of natural-based nanomedicine presents a promising therapeutic approach in the prevention and treatment of neurodegenerative diseases. Several studies have reported on the effects of resveratrol or trans-resveratrol, a polyphenolic molecule, in neurological disorders [222]. Resveratrol has a potentially neuroprotective role in AD, it activates reactive protein kinase C and reduces Aβ toxicity [223]. In a randomized, double-blind trial, patients with mild-to-moderate AD maintained plasma and cerebrospinal fluid Aβ when treated with trans-resveratrol and had reduced markers of inflammation [224]. Another study reported that resveratrol treatment reduces Tau protein levels and inhibits its hyperpolarization in vivo [222]. Curcumin also presents beneficial effects on brain health through Tau inhibition, antioxidation, and enhancing neurogenesis and synaptogenesis [225]. Furthermore, catechins from green tea was found to provide neuroprotection and to stabilize neurodegenerative diseases [226]. Despite the promising effects of natural compounds, the treatment of neurological diseases requires an efficient drug delivery system that overcomes the complexity of the BBB microenvironment. The use of nanomedicine as a therapeutic strategy enhances the safety and efficacy when targeting CNS disorders [227]. Encapsulating curcumin in biodegradable poly lactic-co-glycolic acid nanoparticles enhanced its neuroprotective efficacy through the upregulation of genes linked to cell proliferation and differentiation in vitro, this reversed learning and memory defects in vivo [228]. Similar effects were also observed when resveratrol was encapsulated and coated using polylactic acid and polysorbate-80, respectively [229]. These studies indicate the importance of natural-based nanomedicine for treatment of neurological disorders, the use of nanocarriers enhances the stability and delivery of natural-based molecules to the brain.

## 9. Conclusions

Diabetes is a disorder of dysregulated insulin action or insulin production; the maintenance of energy balance strongly influences the development and progression of disease. Whether this be at the systemic or central level, the mechanisms that maintain energy balance have the potential to strongly influence disease course. In this review, we highlighted those alterations in the neurotransmitter function at synapses and their altered activities can disturb the normal function of the nervous system. These molecules are important in maintaining proper communication between neuronal cells and their microenvironment, centrally and peripherally. Effective neurotransmission in specific brain areas controls appetite and satiety, thus regulating energy homeostasis; several studies justified the association between dysfunctional neurotransmission and diabetes progression. Furthermore, other studies highlighted that risk factors such as insulin resistance and glycemic alterations are the actual contributors to altered neurotransmission in diabetes. Despite that, very limited studies have investigated this concept, and more studies will be required to identify the exact relationship between risk factors for the development of diabetes and dysregulated neurotransmitter function.

At the local tissue level, in the CNS effective neurotransmitter function is maintained by mechanisms that control energy supply to the brain. Several perspectives arose from reviewing the literature in this field. Firstly, considerable technological advances are required to advance this field, for example the development of inducible models that target specific neurotransmitters or their targets. Secondly, once testable candidates have been defined from the relevant models, their application to holistic in vitro or ex vivo screening models will promote selection of translational candidates. Lastly, innovative translational strategies require development, for example the use of intercalating peptides where appropriate, or the development of antisense oligonucleotide therapies that target specific neurotransmitters or neurotransmission machinery. As these areas progress, the repertoire of actionable therapeutic targets will be widened in the areas of metabolic disease, such as diabetes and obesity, as well as in aging or any other condition that is influenced by the neurotransmission process.

## Figures and Tables

**Figure 1 metabolites-13-00384-f001:**
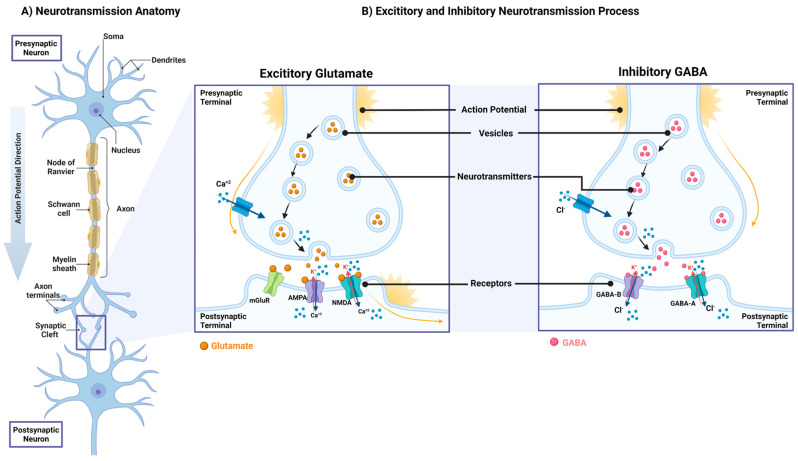
Neurotransmission anatomy and process. (**A**) Neurotransmission anatomy: An action potential moves along the axon of the presynaptic neuron to interact with the dendrites of the postsynaptic neuron creating a synaptic cleft in which the neurotransmission process occurs. (**B**) Excitatory and inhibitory neurotransmission process: Neurotransmitters are stored in vesicles in the presynaptic terminal and are released upon the receipt of an action potential that triggers the release of different ions (i.e., Ca^2+^, Cl^−^, K^+^) and neurotransmitters (i.e., excitatory glutamate and inhibitory GABA) which activates different receptors depending on the synaptic activity. Abbreviations: Ca^2+^: calcium ions; K^+^: potassium ions; mGluR: metabotropic glutamate receptor; AMPA: α-amino-3-hydroxy-5-methyl-4-isoxazole propionic acid receptor; NMDA: N-methyl-D-aspartate receptor; Cl^−^: Chloride ions; GABA_A_ and GABA_b_: γ-Aminobutyric acid receptors (**A**,**B**), respectively. Created with BioRender.com.

**Figure 2 metabolites-13-00384-f002:**
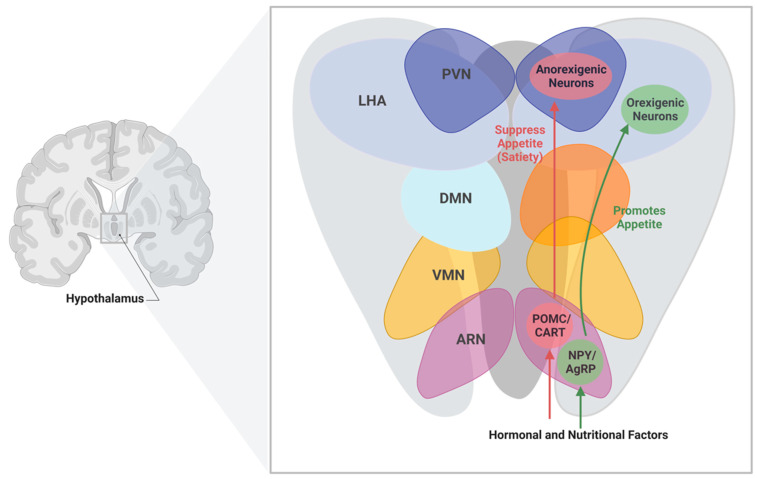
Hypothalamus nuclei associated with controlling energy homeostasis. Schematic representation of the frontal brain section indicating the location of the hypothalamus and the hypothalamic nuclei associated with energy homeostasis. Depending on the signals from the hormonal and nutritional factors, the neuronal connectivity can either suppress or promote appetite through POMC/CART or NPY/AgRP neurons in the ARN, respectively. Abbreviations: PVN: paraventricular nucleus; LHA: lateral hypothalamus area; DMN: dorsal medial nucleus; VMN: ventromedial nucleus; ARN: arcuate nucleus; POMC/CART: proopiomelanocortin/cocaine and amphetamine-regulated transcript; NYP/AgRP: neuropeptide Y/agouti-related protein. Created with BioRender.com.

**Figure 3 metabolites-13-00384-f003:**
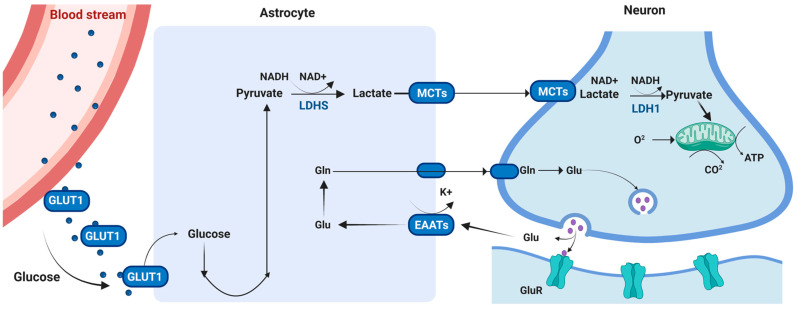
Astrocyte–Neuron Lactate Shuttle (ANLS) Formation. The ANLS is formed during an increase in neuronal energy demand. Astrocytes sense an increase in the glutamatergic excitatory pathway in the neuron and trigger glycolysis by up-taking glucose from the blood stream and converting it into lactate. Lactate is then transported to the neuron through MCTs in which is converted back into pyruvate and then shuttled into the mitochondria as a source of additional energy. Abbreviations: Glut1: glucose transporter protein 1; NADH: nicotinamide adenine dinucleotide with hydrogen; NAD: nicotinamide adenine dinucleotide; LDH: lactate dehydrogenase; MCTs: monocarboxylate transporters; ATP: adenosine 5′-triphosphate; Glu: glutamate; Gln: glutamine; EAATs: excitatory amino acid transporters; GluR: glutamatergic receptor. Created with BioRender.com.

**Table 2 metabolites-13-00384-t002:** Hypothalamus nuclei associated with controlling energy homeostasis [93].

Nuclei	Function
Arcuate Nucleus (ARN)	Regulates energy balance by sensing alterations in hormonal and nutritional factors in the blood stream
Paraventricular Nucleus (PVN)	Maintains whole body energy homeostasis through the regulation of food intake by neuropeptides
Dorsal Medial Nucleus (DMN)	Controls physiological processes and circadian rhythms
Lateral Hypothalamus Area (LHA)	Mediate orexigenic, behavioral, and physiological responses
Ventromedial Nucleus (VMN)	Regulates energy balance and thermogenesis

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
