# Peer review of "Neurotransmitters in Type 2 Diabetes and the Control of Systemic and Central Energy Balance"

_metabolites, 2023, doi:10.3390/metabo13030384_

Round 1
Reviewer 1 Report
The authors present an important aspect of neurotransmitters linked to type II diabetes
I think the initial description of the history of diabetes is superfluous; it should be cut
The influence of lack of physical activity in the pathogenesis of diabetes is well established and should therefore be mentioned
In the causes of the complications, at least the AGEs should be mentioned
The direct effect of glycemic peaks (moderated instead by ketogenic diets) on the excitatory state and, therefore, on neurotransmission should be considered
The authors seem to conclude that the alteration may be the cause of diabetes and/or obesity; in reality, the latter, in particular through insulin resistance and glycemia alteration, causes alteration of neurotransmission, which will surely exacerbate diabetes and/or obesity.
Author Response
Below is the point-by-point response to reviewer 1. Please see attachment for full response to all reviewers.
Reviewer 1
I think the initial description of the history of diabetes is superfluous; it should be cut
[and]
The influence of lack of physical activity in the pathogenesis of diabetes is well established and should therefore be mentioned
We thank the reviewer for these comments. Accordingly, we shortened the introduction and added a brief discussion on the lack of physical activity contributing to energy imbalance and the development of obesity and type-2 diabetes (T2D). The introduction in the revised manuscript is as follows:
“Diabetes and energy balance: Diabetes mellitus is a chronic metabolic disease characterized by hyperglycemia that arises due to insufficient insulin production from the pancreas or from inadequate insulin utilization by the body, or a combination of both (2). Approximately 5% of patients with diabetes suffer from Type-1 Diabetes (T1D), an autoimmune condition in which patients rely on exogenous insulin. The remaining majority have Type-2 Diabetes Mellitus (T2D), characterized by insulin resistance in its early stages. The mechanisms of disease progression in T2D, and other minor frequency forms of diabetes, are under investigation (3). The International Diabetes Federation indicated that 537 million adults globally were living with diabetes in 2021, with an expected 45.8% increase in the number of cases by 2045 (3).
T2D often goes undiagnosed until complications arise, and screening for the disease typically occurs only in obese patients (4). The association between obesity and T2D is strong, yet not essential, as some obese individuals do maintain a healthy metabolism (4,5). Therefore, only some mechanisms of insulin resistance can be associated with obesity. A chronic positive energy balance leads to obesity, and thus generally speaking, increases risk of developing insulin resistance and T2D. Positive energy balance results from either increased energy intake due to high calorie consumption, or decreased energy expenditure due to a lack of physical activity, or a combination of the two. Readjusting energy balance to reduce body weight can improve diabetes. In fact, a 10% reduction in weight has been reported to reverse clinical diagnosis of T2D in some patients with obesity (6).
The basic mechanism for obesity is excess glucose and insulin; glucose stimulates insulin release allowing its utilization by the muscles and storage of the excess as fat. Multiple in vitro and in vivo studies demonstrate that insulin resistance is part of the pathogenesis of T2D as it compromises efficient use of glucose. This also leads to elevated glucagon levels and glucose production by the liver. Persistent high glucose levels are concomitant with the stimulation of fatty acid (FA) release from peripheral storage tissues (i.e., adipose tissue), attempting to provide an alternative energy substrate. However, as the disease progress, FAs fail to be utilized efficiently as an energy substrate, they are stored ectopically, for example in the liver, and this ultimately prevents the compensatory mechanism of increased insulin production (7). Also associated with T2D are increased markers of chronic low-grade systemic inflammation as well as an altered gut microbiome and multi-organ pathologies (8,9).
The role of the brain: Pioneering work by Claude Bernard in the 19th Century linked the brain to systemic glucose homeostasis; in which a needle was used to stimulate a region of the brain in dogs, inducing a temporary diabetic state in the animals (1). In the 1960s, insulin was found to be present in the Central Nervous System (CNS), and rather than being only of pancreatic origin, it was hypothesized and confirmed that the brain can synthesize its own insulin (10,11). Insulin receptors were also found on neurons in many brain regions, with the highest density in the olfactory bulb and hypothalamus (12,13). Functional insulin signaling in the brain via the PI3K/Akt and Ras/Raf/MAP pathways is metabolically protective, neuroprotective, and has a positive effect on neuroplasticity (17). Disruption of insulin action in the brain alters neural and glial cell function at the synaptic level (18,19), and is associated with neurodegenerative and cognitive disorders, as well as psychiatric diseases (20–22). Experimentally, the selective disruption of insulin receptors in the brain also leads to reversible diet-induced obesity and peripheral insulin resistance (23–26) . Multiple studies have also reported an evident link between insulin resistant peripheral tissues and the CNS (e.g., gut-brain axis, liver-brain axis, central leptin resistance) (27,28). Such phenomena are gaining attention for potential roles in systemic insulin resistance that develops in obesity and T2D (28,29).
In this review we discuss how metabolic disturbance, in the form of diabetes, affects neurological functions, with a focus on energy balance and neurotransmission. Neurotransmitters are a class of communication molecules that ensure normal nervous system function, interacting systemically and with various tissue microenvironments. Reviewed below are the ways in which neurotransmitters and neurological function are affected in metabolic disease. To our knowledge, no recent in-depth reviews have addressed how systemic dysmetabolism mechanistically alters neurotransmission and the ways this contributes to increased risk of neurodegenerative diseases in patients with diabetes.”
In the causes of the complications, at least the AGEs should be mentioned
We thank the reviewer for this suggestion. We have now included a short section to discuss the role of advanced glycation end-products (AGE) in the development of neurodegenerative diseases in diabetes. Below is the text incorporated into Section 2 of the revised manuscript:
“From the above studies and other mechanistic investigations, it is clear that impairment of glucose metabolism in the context of diabetes, is a key element in the onset and progression of AD (37,38). In cases of hyperglycemia, reactive oxygen species (ROS) production is elevated through stimulation of the polyol pathway. This leads to formation of advanced glycation end-products (AGE) and causes considerable oxidative stress (39). Accumulation of AGEs contributes to the pathological aspects of neurodegenerative diseases including AD, Parkinson’s and Huntington’s diseases (40). Kong et al., investigated the role of AGEs in AD and diabetes in vivo and found that mice injected with AGEs exhibited symptoms of AD with impaired memory and increased levels of Amyloid Precursor Proteins (APP) and Tau (41). Another study also highlighted that AGE receptor (RAGE), contributes to the decrease of locomotor activity and spatial memory in streptozotocin (STZ)-induced hyperglycemia in mice (42).”
The direct effect of glycemic peaks (moderated instead by ketogenic diets) on the excitatory state and, therefore, on neurotransmission should be considered
We thank the reviewer for this comment. We added a new subsection to the manuscript to discuss the direct effects of glycemic peaks, and glycemic variability in general, on the neurotransmission process. This has been added as Section 7.2 of the revised manuscript:
7.2. Glycemic variability
“Glucose is critical for normal brain metabolism. uncontrolled fluctuations in blood glucose levels stimulate ROS overproduction which makes patients with T2D more susceptible to microvascular complications (179). Recent evidence suggests that fluctuation in glycemic peaks is associated with impairment in cognitive function (180,181). A population based study confirms this further, 5% of the study’s cohort were diagnosed with AD and this was mainly dependent on age and the duration of diabetes (182). Quincozes-Santos et al, showed that fluctuations in glucose levels affected astrocyte function in maintaining cell proliferation and glutamatergic metabolism, inducing cytotoxicity (183). Another study evaluated the effect of fluctuating glycemic peaks on E/I balance in T2D patients and found significant changes in glutamate-GABA neurotransmission which contributes to the pathophysiology of T2D (81).”
The authors seem to conclude that the alteration may be the cause of diabetes and/or obesity; in reality, the latter, in particular through insulin resistance and glycemia alteration, causes alteration of neurotransmission, which will surely exacerbate diabetes and/or obesity.
We thank the reviewer for this insightful comment. Several studies do address the association of diabetes risk factors with neurological outcomes. Indeed, insulin resistance and glycemic variability can contribute to dysregulation of the neurotransmission process. Adding the section on glycemic peaks, above, has strengthened this point. However, beyond what we have described, literature is rather limited in explaining direct neurotransmitter alterations that are caused by diabetes. The Perspectives section of the manuscript has now been updated to reflect areas of future work, including investigations in the direction of causality. Below is the modified text in the Perspectives section of the revised manuscript:
“The current literature was reviewed in the fields of diabetes and neuroscience. The association between the central regulation of energy balance and the development of obesity is clear, as the increased risk of developing T2D in those patients that are obese. Loss of metabolic homeostasis in diabetes, both T2D and T1D, affects multiple organ systems, including the CNS. Indeed, patients with diabetes are at higher risk of developing CNS disorders, as well as peripheral neuropathy. The development of nervous disorders is underpinned by a functional impairment. An important aspect in the function of the nervous system is appropriate neurotransmission function. All the literature cited above indicates that the neurotransmission process is affected by, and can affect the susceptibility to, the development of metabolic diseases. The mechanisms underlying impairments of neurotransmission and CNS function in diabetes are under active investigation, current literature has explained much of the potential mechanisms that link metabolic to neurological dysfunction. Future work in the areas described below will further elucidate molecular and cellular mechanisms underlying the association between diabetes and neurological decline and importantly shed much needed on the directions of causality between metabolic and neurologic disturbance.”

Reviewer 2 Report
1-Please add more in vitro and and animals studies in introduction.
2.what is the suggestion of this study for future works?
3.Please discuss the use of natural based nanomedicine targeting neural system.
4.Pleas discuss the role of gap junctions on neural system and their effect on diabetes.
5.Please discuss the role of cross talk between mitochondria and Neurotransmitters on diabetes.
6.There are many studies investigating the importance of topic , Please add these references to your discussion part of the manuscript and compare and bold your study novelty :
-DOI: 10.1016/j.cellsig.2019.03.010
-DOI: 10.1155/2021/4946711
Author Response
Below is the point-by-point response to reviewer 2. Please see attachment for full response to all reviewers.
Reviewer 2
1-Please add more in vitro and animals studies in introduction.
We thank the reviewer for this comment, we have now rewritten the introduction and included studies that have drawn mechanisms from in vitro and in vivo studies. The updated introduction in the revised manuscript is as follows:
“Diabetes and energy balance: Diabetes mellitus is a chronic metabolic disease characterized by hyperglycemia that arises due to insufficient insulin production from the pancreas or from inadequate insulin utilization by the body, or a combination of both (2). Approximately 5% of patients with diabetes suffer from Type-1 Diabetes (T1D), an autoimmune condition in which patients rely on exogenous insulin. The remaining majority have Type-2 Diabetes Mellitus (T2D), characterized by insulin resistance in its early stages. The mechanisms of disease progression in T2D, and other minor frequency forms of diabetes, are under investigation (3). The International Diabetes Federation indicated that 537 million adults globally were living with diabetes in 2021, with an expected 45.8% increase in the number of cases by 2045 (3).
T2D often goes undiagnosed until complications arise, and screening for the disease typically occurs only in obese patients (4). The association between obesity and T2D is strong, yet not essential, as some obese individuals do maintain a healthy metabolism (4,5). Therefore, only some mechanisms of insulin resistance can be associated with obesity. A chronic positive energy balance leads to obesity, and thus generally speaking, increases risk of developing insulin resistance and T2D. Positive energy balance results from either increased energy intake due to high calorie consumption, or decreased energy expenditure due to a lack of physical activity, or a combination of the two. Readjusting energy balance to reduce body weight can improve diabetes. In fact, a 10% reduction in weight has been reported to reverse clinical diagnosis of T2D in some patients with obesity (6).
The basic mechanism for obesity is excess glucose and insulin; glucose stimulates insulin release allowing its utilization by the muscles and storage of the excess as fat. Multiple in vitro and in vivo studies demonstrate that insulin resistance is part of the pathogenesis of T2D as it compromises efficient use of glucose. This also leads to elevated glucagon levels and glucose production by the liver. Persistent high glucose levels are concomitant with the stimulation of fatty acid (FA) release from peripheral storage tissues (i.e., adipose tissue), attempting to provide an alternative energy substrate. However, as the disease progress, FAs fail to be utilized efficiently as an energy substrate, they are stored ectopically, for example in the liver, and this ultimately prevents the compensatory mechanism of increased insulin production (7). Also associated with T2D are increased markers of chronic low-grade systemic inflammation as well as an altered gut microbiome and multi-organ pathologies (8,9).
The role of the brain: Pioneering work by Claude Bernard in the 19th Century linked the brain to systemic glucose homeostasis; in which a needle was used to stimulate a region of the brain in dogs, inducing a temporary diabetic state in the animals (1). In the 1960s, insulin was found to be present in the Central Nervous System (CNS), and rather than being only of pancreatic origin, it was hypothesized and confirmed that the brain can synthesize its own insulin (10,11). Insulin receptors were also found on neurons in many brain regions, with the highest density in the olfactory bulb and hypothalamus (12,13). Functional insulin signaling in the brain via the PI3K/Akt and Ras/Raf/MAP pathways is metabolically protective, neuroprotective, and has a positive effect on neuroplasticity (17). Disruption of insulin action in the brain alters neural and glial cell function at the synaptic level (18,19), and is associated with neurodegenerative and cognitive disorders, as well as psychiatric diseases (20–22). Experimentally, the selective disruption of insulin receptors in the brain also leads to reversible diet-induced obesity and peripheral insulin resistance (23–26) . Multiple studies have also reported an evident link between insulin resistant peripheral tissues and the CNS (e.g., gut-brain axis, liver-brain axis, central leptin resistance) (27,28). Such phenomena are gaining attention for potential roles in systemic insulin resistance that develops in obesity and T2D (28,29).
In this review we discuss how metabolic disturbance, in the form of diabetes, affects neurological functions, with a focus on energy balance and neurotransmission. Neurotransmitters are a class of communication molecules that ensure normal nervous system function, interacting systemically and with various tissue microenvironments. Reviewed below are the ways in which neurotransmitters and neurological function are affected in metabolic disease. To our knowledge, no recent in-depth reviews have addressed how systemic dysmetabolism mechanistically alters neurotransmission and the ways this contributes to increased risk of neurodegenerative diseases in patients with diabetes.”
2.what is the suggestion of this study for future works?
We thank the reviewer for this insightful comment. In the revised manuscript we have separated and modified the perspectives section and the conclusion sections. We have made use of the perspectives section to clearly indicate priority areas of future work in the fields of neurological disease and diabetes research. Accordingly, the new section entitled “8. perspectives and future work in the field” reads as follows:
“8. Perspectives and future work in the field
We reviewed the current literature in the fields of diabetes and in neuroscience. The association between the central regulation of energy balance and the development of obesity are clear, as is the increased risk of developing T2D in those patients that are obese. Loss of metabolic homeostasis in diabetes, both T2D and T1D, affects multiple organ systems, including the CNS. Indeed, patients with diabetes are at higher risk of developing CNS disorders, as well as peripheral neuropathy. The development of nervous disorders is underpinned by a functional impairment. An important aspect in the function of the nervous system is appropriate signal transduction by way of neurotransmission. All the literature cited above indicates both that the neurotransmission process is affected by, and can affect susceptibility to, the development of metabolic diseases. The mechanisms underlying impairments of neurotransmission and CNS function in diabetes are under active investigation, current literature has explained much of the potential mechanisms that link metabolic to neurological dysfunction. Future work in the areas described below will further elucidate molecular and cellular mechanisms underlying the association between diabetes and neurological decline and importantly shed much needed on the directions of causality between metabolic and neurologic disturbance.
8.1 Emerging role of inflammation in neurotransmitter function
The nervous and immune systems are tightly intertwined as they modulate each other through sophisticated bidirectional crosstalk. Specifically, the brain is a highly immunologically active organ that hosts its own immune cells and allows others to circulate through its fluid filled borders in the meninges, the protective membranes that cover the CNS (198). This is evident from multiple studies, for example, knocking out immune cells from mice led to accelerated neurodegeneration that was accompanied by gliosis; while restoring these cells modulated the trophic/cytotoxic balance of glial cells and slowed disease progression (199). Another study showed that patients with AD had a significant rise in the number of T-cells in the cerebrospinal fluid suggesting that they might have a role in AD progression (200). On the contrary, instead of its neuroprotective role, the immune system can also cause cumulative damage to neurons due to chronic inflammatory reactions (201).
8.2. Neuroinflammation
Neuroinflammation refers to the accumulation of glial cells in the central nervous system in response to inflammation (201). This occurs when astrocytes and microglia are activated immediately after injury leading to the secretion of proinflammatory cytokines (e.g., IL-1β, TNFα), cytotoxic compounds and ROS, triggering neuronal death (201). GABA and glutamate receptor expression is altered under neuroinflammatory conditions causing impaired spatial learning, motor function and cognitive decline (202). Experimentally induced inflammation through administration of poly(I:C) during gestation, dysregulated prefrontal GABAergic expression and led to the development of long-term neuropsychiatric disorders in animal models (203). Moreover, GABA suppresses the reactive response between astrocytes and microglia to inflammatory stimuli and reduces the release of TNFα and IL-6; while an increase in TNFα levels downregulates the inhibitory synaptic strength of GABAergic neurotransmission (204,205). Another study also looked at the effect of parthenolide, an NF-kB inhibitor, on diabetic rat models and found that the rats exhibited enhanced cognition, reduced anxiety-like behavior, and decreased TNFα and IL-6 levels in the cortex and hippocampus; which was associated with the loss of GABA and glutamate homeostasis (206).
8.3. Hypothalamic Inflammation
Since the hypothalamus is a major site for appetite regulation and energy balance, hypothalamic inflammation is often suggested as a cause for dysregulation of feeding behavior and body weight (207). Obesity can, in fact, be considered as a hypothalamic inflammatory disorder resulting from the accumulation of high-fat and high glycemic index products that accumulate as ectopic fat. This eventually leads to increased BBB permeability, tanycytes damage and activation of microglia and astrocytes pro-inflammatory pathways; that induces the secretion of cytokines such as IL-1β and TNFα alongside ROS production in the hypothalamus (208). This is evident in patients with diabetes and obesity as a result of dysregulated energy homeostasis (208). Moreover, obese insulin resistant rat models of AD exhibited elevated levels of pro-inflammatory cytokines and chemokines in the hypothalamus and other brain regions suggesting a correlation between AD and hypothalamic inflammation (209).
8.4. Hypothalamic regulation of energy homeostasis: the roles of sleep and thermogenesis
Thermal and sleep regulation are also associated with the hypothalamus’ role in maintaining energy homeostasis. Previous studies indicate that AgRP and POMC stimulation promotes wakefulness and reduces sleep fragmentation, respectively (210). The hypothalamus contains a sleep and thermal regulation center known as the ventrolateral preoptic (VLPO) area. The VLPO consists of a large population of GABAergic neurons that send signals to other brain regions to regulate sleep (210). Disruption of GABAergic neurons in the LHA dysregulates sleep and induces vigorous eating habits. A study looked at GABAergic neurons in the DMN region of the mouse hypothalamus and found that these neurons play a role in sleep-wake changes elicited by exposure to warmth in mice, indicative of their role in maintaining sleep and energy homeostasis (211).
8.5. Therapeutic approaches for neurological disorders
The use of natural based nanomedicine presents a promising therapeutic approach in the prevention and treatment of neurodegenerative diseases. Several studies reported on the effects of resveratrol or trans-resveratrol, a polyphenolic molecule, in neurological disorders (212). Resveratrol has a potentially neuroprotective role in AD, it activates reactive protein kinase C and reduces Aβ toxicity (213). In a randomized, double-blind trial, patients with mild-to-moderate AD maintained plasma and cerebrospinal fluid Aβ when treated with trans-resveratrol and had reduced markers of inflammation (214). Another study reported that resveratrol treatment reduces Tau protein levels and inhibits its hyperpolarization in vivo (212). Curcumin also presents beneficial effects on brain health through Tau inhibition, antioxidation, and enhancing neurogenesis and synaptogenesis (215). Catechins from green tea have been found to provide neuroprotection and to stabilize neurodegenerative diseases (216). Despite the promising effects of natural compounds, the treatment of neurological diseases requires an efficient drug delivery system that overcomes the complexity of the BBB microenvironment. The use of nanomedicine as a therapeutic strategy enhances the safety and efficacy when targeting CNS disorders (217). Encapsulating curcumin in biodegradable poly-lacticco-glycolic acid nanoparticles enhanced its neuroprotective efficacy through the upregulation of genes linked to cell proliferation and differentiation in vitro, this reversed learning and memory defects in vivo (218). Similar effects were also observed when resveratrol was encapsulated and coated using polylactic acid and polysorbate-80 respectively (219). These studies indicate the importance of natural based nanomedicine for treatment of neurological disorders, the use of nanocarriers enhances the stability and delivery of natural based molecules to the brain.”
3.Please discuss the use of natural based nanomedicine targeting neural system.
We thank the reviewer for this suggestion. The review was expanded to include studies on natural based nanomedicine in targeting the nervous system. Below are the relevant studies cited and the text incorporated into section 8.5: Therapeutic approaches for neurological disorders (page 14):
8.5. Therapeutic approaches for neurological disorders
The use of natural based nanomedicine presents a promising therapeutic approach in the prevention and treatment of neurodegenerative diseases. Several studies reported on the effects of resveratrol or trans-resveratrol, a polyphenolic molecule, in neurological disorders (215). Resveratrol has a potentially neuroprotective role in AD, it activates reactive protein kinase C and reduces Aβ toxicity (216). In a randomized, double-blind trial, patients with mild-to-moderate AD maintained plasma and cerebrospinal fluid Aβ when treated with trans-resveratrol and had reduced markers of inflammation (217). Another study reported that resveratrol treatment reduces Tau protein levels and inhibits its hyperpolarization in vivo (215). Curcumin also presents beneficial effects on brain health through Tau inhibition, antioxidation, and enhancing neurogenesis and synaptogenesis (218). Furthermore, catechins from green tea was found to provide neuroprotection and to stabilize neurodegenerative diseases (219). Despite the promising effects of natural compounds, the treatment of neurological diseases requires an efficient drug delivery system that overcomes the complexity of the BBB microenvironment. The use of nanomedicine as a therapeutic strategy enhances the safety and efficacy when targeting CNS disorders (220). Encapsulating curcumin in biodegradable poly-lacticco-glycolic acid nanoparticles enhanced its neuroprotective efficacy through the upregulation of genes linked to cell proliferation and differentiation in vitro, this reversed learning and memory defects in vivo (221). Similar effects were also observed when resveratrol was encapsulated and coated using polylactic acid and polysorbate-80 respectively (222). These studies indicate the importance of natural based nanomedicine for treatment of neurological disorders, the use of nanocarriers enhances the stability and delivery of natural based molecules to the brain.
4.Please discuss the role of gap junctions on neural system and their effect on diabetes.
We thank the reviewer for this suggestion. The review was expanded to include studies on the role of gap junctions in the CNS and how it is affected by diabetes. Below are the relevant studies cited and the text incorporated into the following sections:
Neurotransmission also occurs through electrical synapses that contain intercellular aggregate channels known as Gap Junctions (GJs) allowing electrical and metabolic communication between adjacent neurons. GJs are formed by hemichannels from each side of the synapse which are composed of transmembrane proteins called connexins that allow the transfer of ions and small molecules between neurons (72,73). Cx36 is the most abundant connexin type in neurons and it forms most of the electrical synapses in the CNS (74). Connexin GJs play a homeostatic role in CNS physiology. This includes synaptogenesis, neuronal differentiation and circuit formation and maturation (73). GJs also regulate neural activity oscillations (i.e. maintaining a synchronized excitatory and inhibitory electrical activity) that allow robust communication between neuronal assemblies. Alterations in connexin GJ activities can impact their expression and function leading to the progression of neurodegenerative diseases, including AD and Parkinson’s disease and epilepsy (75–78).
[and]
Furthermore, hyperglycemic conditions lead to GJ impairment that disrupts astrocyte-neuron communication leading to changes in brain function (85). Head et al. reported that loss of Cx36 GJs disrupted glucose homeostasis through the alteration of oscillating insulin levels in mice (86). This was further confirmed in patients with diabetes, as they exhibited disruption in Cx36 GJ permeability and Ca2+ electrical activity. Treatment with Modafinil restored Cx36 GJ function, maintained cell viability and and protected against β-cell dysfunction (87).
5.Please discuss the role of cross talk between mitochondria and Neurotransmitters on diabetes.
We thank the reviewer for this suggestion. The review was expanded to include studies on the association between mitochondria and neurotransmitters in the context of diabetes. Below are the relevant studies cited and the text incorporated into section 6
Oxidative damage and mitochondrial dysfunction are key pathological processes that mediate neurodegeneration in diabetes. Abnormalities in mitochondrial function, structure and connectivity can impact behavior and contribute to obesity (166). Moreira et al., showed that mice under HFD exhibited reduced ATP production, altered mitochondrial dynamics and fragmentation in the hippocampus (167). HFD also led to deficits in electron transport chain function and in oxidative phosphorylation, with reduced expression of mitochondria associated proteins such as PGC-1a and the mitochondrial transcription factor A (TFAM) in vivo (168). Moreover, the link between insulin resistance and behavioral disorders such as anxiety and depression are associated with dysregulation of mitochondrial function and monoamine neurotransmitter homeostasis. This was demonstrated in neuron-specific insulin receptor knock-out mice that had elevated dopamine clearance and reduced mitochondrial oxidative activity (26). Another study induced a diabetes-like phenotype in MitoPark mouse model, which exhibits a Parkinsonian phenotype and mitochondrial dysfunction in dopamine neurons. This was observed in vivo and similar results were observed ex vivo, collectively indicating that mitochondrial dysfunction is associated with the progression of Parkinson’s disease in the context of diabetes (169).
6.There are many studies investigating the importance of topic , Please add these references to your discussion part of the manuscript and compare and bold your study novelty :
-DOI: 10.1016/j.cellsig.2019.03.010
-DOI: 10.1155/2021/4946711
We thank the reviewer for this suggestion. The first reference was added to section 3: Neurotransmitters as it was linked to the neuronal gap junction concept. The aspect on the link between mitochondria and neurotransmission in diabetes was addressed in point 5, and we consider the recommended reference on the role of polyphenol compounds in modulating signaling pathways of mitochondrial biogenesis to be beyond the scope of the current review. Here we mainly explain the role of neurotransmitters in regulating energy balance in diabetes and how dysregulation in neurotransmission lead to neurodegenerative diseases and disorders.

Reviewer 3 Report
Title: Neurotransmitters in type-2 diabetes and the control of systemic and central energy balance
The literature review is informative, well-written, and generally well-structure. There is a need for such studies because of the alarming rise in Obesity, insulin resistance, DM, and neurodegenerative and cognitive disorders, prevalence across the globe, as well as the need for alternative remedies to tackle these disorders. I have a few concerns though.
1. Introduction:
- The author used very old literature or articles; I strongly recommend updating/substituting with more recent references throughout the manuscript.
-This section looks promising. However, authors should clarify the novelty of this article in the ‘Introduction’ and ‘Conclusion’ section.
2. Discussion: How is this article more informative than the previously published ones? Justify it.
3. Conclusion: This section looks okay. However, I would suggest to summarize in brief rather than descriptive.
4. Overall, it looks promising.
Author Response
Below is the point-by-point response to reviewer 3. Please see attachment for full response to all reviewers.
Reviewer 3
- Introduction:
The author used very old literature or articles; I strongly recommend updating/substituting with more recent references throughout the manuscript.
We thank the reviewer for this suggestion, we have updated the revised manuscript with more recent.
This section looks promising. However, authors should clarify the novelty of this article in the ‘Introduction’ and ‘Conclusion’ section
[and]
- Discussion:How is this article more informative than the previously published ones? Justify it.
[and]
- Conclusion: This section looks okay. However, I would suggest to summarize in brief rather than descriptive.
We thank the reviewer for these comments. We have rewritten both the introduction and the conclusion sections to emphasize novelty and the need for such a review to be conducted. We have also separated the section on perspectives and developed it as a section to discuss future work in the field. We believe the rewritten introduction, conclusion and perspectives sections all add great value in justifying review and in discussing contents of the revised manuscript. These sections are updated as follows:
Introduction :
“Diabetes and energy balance: Diabetes mellitus is a chronic metabolic disease characterized by hyperglycemia that arises due to insufficient insulin production from the pancreas or from inadequate insulin utilization by the body, or a combination of both (2). Approximately 5% of patients with diabetes suffer from Type-1 Diabetes (T1D), an autoimmune condition in which patients rely on exogenous insulin. The remaining majority have Type-2 Diabetes Mellitus (T2D), characterized by insulin resistance in its early stages. The mechanisms of disease progression in T2D, and other minor frequency forms of diabetes, are under investigation (3). The International Diabetes Federation indicated that 537 million adults globally were living with diabetes in 2021, with an expected 45.8% increase in the number of cases by 2045 (3).
T2D often goes undiagnosed until complications arise, and screening for the disease typically occurs only in obese patients (4). The association between obesity and T2D is strong, yet not essential, as some obese individuals do maintain a healthy metabolism (4,5). Therefore, only some mechanisms of insulin resistance can be associated with obesity. A chronic positive energy balance leads to obesity, and thus generally speaking, increases risk of developing insulin resistance and T2D. Positive energy balance results from either increased energy intake due to high calorie consumption, or decreased energy expenditure due to a lack of physical activity, or a combination of the two. Readjusting energy balance to reduce body weight can improve diabetes. In fact, a 10% reduction in weight has been reported to reverse clinical diagnosis of T2D in some patients with obesity (6).
The basic mechanism for obesity is excess glucose and insulin; glucose stimulates insulin release allowing its utilization by the muscles and storage of the excess as fat. Multiple in vitro and in vivo studies demonstrate that insulin resistance is part of the pathogenesis of T2D as it compromises efficient use of glucose. This also leads to elevated glucagon levels and glucose production by the liver. Persistent high glucose levels are concomitant with the stimulation of fatty acid (FA) release from peripheral storage tissues (i.e., adipose tissue), attempting to provide an alternative energy substrate. However, as the disease progress, FAs fail to be utilized efficiently as an energy substrate, they are stored ectopically, for example in the liver, and this ultimately prevents the compensatory mechanism of increased insulin production (7). Also associated with T2D are increased markers of chronic low-grade systemic inflammation as well as an altered gut microbiome and multi-organ pathologies (8,9).
The role of the brain: Pioneering work by Claude Bernard in the 19th Century linked the brain to systemic glucose homeostasis; in which a needle was used to stimulate a region of the brain in dogs, inducing a temporary diabetic state in the animals (1). In the 1960s, insulin was found to be present in the Central Nervous System (CNS), and rather than being only of pancreatic origin, it was hypothesized and confirmed that the brain can synthesize its own insulin (10,11). Insulin receptors were also found on neurons in many brain regions, with the highest density in the olfactory bulb and hypothalamus (12,13). Functional insulin signaling in the brain via the PI3K/Akt and Ras/Raf/MAP pathways is metabolically protective, neuroprotective, and has a positive effect on neuroplasticity (17). Disruption of insulin action in the brain alters neural and glial cell function at the synaptic level (18,19), and is associated with neurodegenerative and cognitive disorders, as well as psychiatric diseases (20–22). Experimentally, the selective disruption of insulin receptors in the brain also leads to reversible diet-induced obesity and peripheral insulin resistance (23–26) . Multiple studies have also reported an evident link between insulin resistant peripheral tissues and the CNS (e.g., gut-brain axis, liver-brain axis, central leptin resistance) (27,28). Such phenomena are gaining attention for potential roles in systemic insulin resistance that develops in obesity and T2D (28,29).
In this review we discuss how metabolic disturbance, in the form of diabetes, affects neurological functions, with a focus on energy balance and neurotransmission. Neurotransmitters are a class of communication molecules that ensure normal nervous system function, interacting systemically and with various tissue microenvironments. Reviewed below are the ways in which neurotransmitters and neurological function are affected in metabolic disease. To our knowledge, no recent in-depth reviews have addressed how systemic dysmetabolism mechanistically alters neurotransmission and the ways this contributes to increased risk of neurodegenerative diseases in patients with diabetes.”
Perspectives and future work in the field:
“We reviewed the current literature in the fields of diabetes and in neuroscience. The association between the central regulation of energy balance and the development of obesity are clear, as is the increased risk of developing T2D in those patients that are obese. Loss of metabolic homeostasis in diabetes, both T2D and T1D, affects multiple organ systems, including the CNS. Indeed, patients with diabetes are at higher risk of developing CNS disorders, as well as peripheral neuropathy. The development of nervous disorders is underpinned by a functional impairment. An important aspect in the function of the nervous system is appropriate signal transduction by way of neurotransmission. All the literature cited above indicates both that the neurotransmission process is affected by, and can affect susceptibility to, the development of metabolic diseases. The mechanisms underlying impairments of neurotransmission and CNS function in diabetes are under active investigation, current literature has explained much of the potential mechanisms that link metabolic to neurological dysfunction. Future work in the areas described below will further elucidate molecular and cellular mechanisms underlying the association between diabetes and neurological decline and importantly shed much needed on the directions of causality between metabolic and neurologic disturbance.
8.1 Emerging role of inflammation in neurotransmitter function
The nervous and immune systems are tightly intertwined as they modulate each other through sophisticated bidirectional crosstalk. Specifically, the brain is a highly immunologically active organ that hosts its own immune cells and allows others to circulate through its fluid filled borders in the meninges, the protective membranes that cover the CNS (198). This is evident from multiple studies, for example, knocking out immune cells from mice led to accelerated neurodegeneration that was accompanied by gliosis; while restoring these cells modulated the trophic/cytotoxic balance of glial cells and slowed disease progression (199). Another study showed that patients with AD had a significant rise in the number of T-cells in the cerebrospinal fluid suggesting that they might have a role in AD progression (200). On the contrary, instead of its neuroprotective role, the immune system can also cause cumulative damage to neurons due to chronic inflammatory reactions (201).
8.2. Neuroinflammation
Neuroinflammation refers to the accumulation of glial cells in the central nervous system in response to inflammation (201). This occurs when astrocytes and microglia are activated immediately after injury leading to the secretion of proinflammatory cytokines (e.g., IL-1β, TNFα), cytotoxic compounds and ROS, triggering neuronal death (201). GABA and glutamate receptor expression is altered under neuroinflammatory conditions causing impaired spatial learning, motor function and cognitive decline (202). Experimentally induced inflammation through administration of poly(I:C) during gestation, dysregulated prefrontal GABAergic expression and led to the development of long-term neuropsychiatric disorders in animal models (203). Moreover, GABA suppresses the reactive response between astrocytes and microglia to inflammatory stimuli and reduces the release of TNFα and IL-6; while an increase in TNFα levels downregulates the inhibitory synaptic strength of GABAergic neurotransmission (204,205). Another study also looked at the effect of parthenolide, an NF-kB inhibitor, on diabetic rat models and found that the rats exhibited enhanced cognition, reduced anxiety-like behavior, and decreased TNFα and IL-6 levels in the cortex and hippocampus; which was associated with the loss of GABA and glutamate homeostasis (206).
8.3. Hypothalamic Inflammation
Since the hypothalamus is a major site for appetite regulation and energy balance, hypothalamic inflammation is often suggested as a cause for dysregulation of feeding behavior and body weight (207). Obesity can, in fact, be considered as a hypothalamic inflammatory disorder resulting from the accumulation of high-fat and high glycemic index products that accumulate as ectopic fat. This eventually leads to increased BBB permeability, tanycytes damage and activation of microglia and astrocytes pro-inflammatory pathways; that induces the secretion of cytokines such as IL-1β and TNFα alongside ROS production in the hypothalamus (208). This is evident in patients with diabetes and obesity as a result of dysregulated energy homeostasis (208). Moreover, obese insulin resistant rat models of AD exhibited elevated levels of pro-inflammatory cytokines and chemokines in the hypothalamus and other brain regions suggesting a correlation between AD and hypothalamic inflammation (209).
8.4. Hypothalamic regulation of energy homeostasis: the roles of sleep and thermogenesis
Thermal and sleep regulation are also associated with the hypothalamus’ role in maintaining energy homeostasis. Previous studies indicate that AgRP and POMC stimulation promotes wakefulness and reduces sleep fragmentation, respectively (210). The hypothalamus contains a sleep and thermal regulation center known as the ventrolateral preoptic (VLPO) area. The VLPO consists of a large population of GABAergic neurons that send signals to other brain regions to regulate sleep (210). Disruption of GABAergic neurons in the LHA dysregulates sleep and induces vigorous eating habits. A study looked at GABAergic neurons in the DMN region of the mouse hypothalamus and found that these neurons play a role in sleep-wake changes elicited by exposure to warmth in mice, indicative of their role in maintaining sleep and energy homeostasis (211).
8.5. Therapeutic approaches for neurological disorders
Use of natural based nanomedicine presents a promising therapeutic approach in the prevention and treatment of neurodegenerative diseases. Several studies reported on the effects of resveratrol or trans-resveratrol, a polyphenolic molecule, in neurological disorders (212). Resveratrol has a potentially neuroprotective role in AD, it activates reactive protein kinase C and reduces Aβ toxicity (213). In a randomized, double-blind trial, patients with mild-to-moderate AD maintained plasma and cerebrospinal fluid Aβ when treated with trans-resveratrol and had reduced markers of inflammation (214). Another study reported that resveratrol treatment reduces Tau protein levels and inhibits its hyperpolarization in vivo (212). Curcumin also presents beneficial effects on brain health through Tau inhibition, antioxidation, and enhancing neurogenesis and synaptogenesis (215). Catechins from green tea have been found to provide neuroprotection and to stabilize neurodegenerative diseases (216). Despite the promising effects of natural compounds, the treatment of neurological diseases requires an efficient drug delivery system that overcomes the complexity of the BBB microenvironment. The use of nanomedicine as a therapeutic strategy enhances the safety and efficacy when targeting CNS disorders (217). Encapsulating curcumin in biodegradable poly-lacticco-glycolic acid nanoparticles enhanced its neuroprotective efficacy through the upregulation of genes linked to cell proliferation and differentiation in vitro, this reversed learning and memory defects in vivo (218). Similar effects were also observed when resveratrol was encapsulated and coated using polylactic acid and polysorbate-80 respectively (219). These studies indicate the importance of natural based nanomedicine for treatment of neurological disorders, the use of nanocarriers enhances the stability and delivery of natural based molecules to the brain.s”
Conclusion :
“Diabetes is a disorder of dysregulated insulin action or insulin production; the maintenance of energy balance strongly influences the development and progression of disease. Whether this be at the systemic or central level, the mechanisms that maintain energy balance have the potential to strongly influence disease course. In this review, we highlighted those alterations in neurotransmitter function at synapses and their altered activities can disturb the normal function of the nervous system. These molecules are important in maintaining proper communication between neuronal cells and their microenvironment, centrally and peripherally. Effective neurotransmission in specific brain areas controls appetite and satiety thus regulating energy homeostasis; several studies justified the association between dysfunctional neurotransmission and diabetes progression. Furthermore, other studies highlighted that risk factors such as insulin resistance and glycemic alterations are the actual contributors to altered neurotransmission in diabetes. Despite that, very limited studies investigated this concept, and more studies will be required to identify the exact relationship between risk factors for the development of diabetes and dysregulated neurotransmitter function.
At the local tissue level, in the CNS effective neurotransmitter function is maintained by mechanisms that control energy supply to the brain. Several perspectives have arisen from reviewing the literature in this field. Firstly, considerable technological advances are required to advance this field, for example the development of inducible models that target specific neurotransmitters or their targets. Secondly, once testable candidates have been defined from the relevant models, their application to holistic in vitro or ex vivo screening models will promote selection of translational candidates. Lastly, innovative translational strategies require development, for example the use of intercalating peptides where appropriate, or the development of antisense oligonucleotide therapies that target specific neurotransmitters or neurotransmission machinery. As these areas progress, the repertoire of actionable therapeutic targets will be widened in the areas of metabolic disease, such as diabetes and obesity, as well as in aging or any other condition that is influenced by the neurotransmission process.”

Round 2
Reviewer 1 Report
The authors improved the manuscript enough even some point should be improved, in particular the influence of ketogenic diet and consequently of ketone bodies.
Author Response
We thank the reviewer for this suggestion. We added a paragraph dedicated to ketogenic diet and ketone bodies. The below text was incorporated into the review:
“Ketone bodies can also act as an energy substrate, especially in the case of diabetes or under conditions of starvation or fasting when glucose levels are low (144,149). Physiologically, ketone bodies are generated in the liver when metabolism switches from carbohydrates to fats. Such conditions can be induced by a ketogenic diet (KD), which is a high fat and low carbohydrate diet (150). Once ketone bodies cross the Blood-Brain-Barrier (BBB), they enhance mitochondrial function, ATP generation, synaptic plasticity and the myelination process at the early stages of brain development, making them a potential therapeutic target for neurodegenerative diseases (144,150). Administrating KD in patients with AD showed significant improvements in cognitive and executive functions (151), Likewise, patients with Parkinson’s also had enhanced non-motor functions and cognition (152). Furthermore, studies demonstrated that ketone bodies can reduce glutamate excitatory neurotransmission effect and the firing rates of neurons by opening potassium ATP channels and activating GABA receptors providing a therapeutic mechanism for AD and epilepsy (153,154). Hypometabolism is often accelerated in the context of insulin resistance which destabilizes the brain network, triggering diabetes-induced dementia (150,155). Some studies highlight the role of KD in improving insulin sensitivity and glycemic control (150,156). Paradoxically, patients with diabetes are more prone to diabetic ketoacidosis, which can lead to comas or death. Diabetic ketoacidosis occurs when the body does not have enough insulin to trigger the use of glucose, lipids are then used to compensate, resulting in production of ketones and their build up in blood. Despite potential benefits of ketone bodies in neurodegenerative disease, prolonged exposure to them in the hypothalamus, can further dysregulate insulin secretion and energy homeostasis, leading to hyperphagia and obesity (157,158).”
Please see attachment for cover letter and revised manuscript.
